# Facebook addiction and affected academic performance among Ethiopian university students: A cross-sectional study

**Aman Dule** [1]*, **Zakir Abdu** [1], **Mohammedamin Hajure** [1], **Mustefa Mohammedhussein** [2], **Million Girma** [1‡], **Wubishet Gezimu** [3‡], **Abdissa Duguma** [3‡]

1 Department of Psychiatry, Collage of Health Sciences, Mettu University, Mettu, Oromia, Ethiopia,
2 Department of Psychiatry, School of Health Sciences, Madda Walabu University, Goba, Oromia, Ethiopia,
3 Department of Nursing, Collage of Health Sciences, Mettu University, Mettu, Oromia, Ethiopia

☯ These authors contributed equally to this work.
‡ MG, WG and AD also contributed to this work equally.
* amandule1993@gmail.com

**Data Availability Statement:** All relevant data are within the paper and its Supporting Information files.

## Abstract

Addiction is an extreme craving for and commitment to something, physically or psychologically. Currently, addiction to social media is the main emerging technology addiction, especially among the young generation. The main aim of the current study was to evaluate the status of Facebook addiction and its relation to academic performance and other correlates among university students. A cross-sectional study was conducted among 422 students from December 1–30, 2021, and Facebook addiction was examined with the Bergen Facebook Addiction Scale (BFAS). The Rosenberg Self-Esteem Scale (RSES), Hospital Anxiety and Depression Scale (HADS), and Study Habit Questionnaire (SHQ) were employed to assess self-esteem, anxiety and depression symptoms, and study habits, respectively. Systematic random sampling was used to recruit the subjects, and the data were analyzed by SPSS version 23.0. Statistics such as percentages, frequencies, mean ± SD, and mean differences were calculated. Multiple regression analysis was performed, and all the required assumptions were checked. The statistical significance was declared at a p-value < 0.05 and a 95% CI. Results revealed that, the mean age of the students was 23.62 (SD = ±1.79) and 51.6% of the participants were male. The majority of the participants were addicted to Facebook, and Facebook addiction was positively linked with factors like lower academic achievements and the symptoms of anxiety and depression. In conclusion, Facebook addiction was found to be higher among study participants, and it is negatively affecting their academic performances. Similarly, it was associated with affected mental well-being and reduced self-esteem. It is better for the legislative body of the university to put firm policies in place for promoting safe use and reducing the detrimental effects of this problem among students.

**Funding:** The author(s) received no specific funding for this work.

**Competing interests:** The authors have declared that no competing interests exist.

## Introduction

In a medical context, addiction is defined as an extreme craving for and commitment to something, either physically or psychologically [1]. In the current era, internet addiction is the main emerging technology addiction [2]. Serving as ways of connection among people, addiction to the internet and social media has become a pandemic globally [3].

Among available social media, Facebook has become the chief means of interaction, especially among university students [4]. Despite its usage as the bridge of connection, it is considered an emerging challenge in different aspects, especially among the young generation [5] and university students, partly because of its heavy and aimless usage [6]. As its users are currently increasing, controlling the detrimental effects of Facebook is becoming more challenging, specifically in developing countries with unconfirmed regulatory policies [7].

As the studies revealed, extended use of Facebook has led to poor academic performance, bullying activities, decreased face-to-face contact, sleep disruptions, and mental health disturbances [8, 9]. Log-in-related distractions such as uploading, commenting, and chatting with friends are leading to procrastination of learning activities [8] and sinking academic success among university students currently [10].

The previous study revealed that too much use of Facebook was related to a lower grade point average (GPA) and disturbances in daily routine activities [9]. It has been reported that students who are more addicted to Facebook have poor study habits and lower academic achievements [11]. Similarly, those adolescents who were addicted to Facebook demonstrated poor study habits, which resulted in deprived academic performances [12]. Another study has shown that students with high Facebook addiction had disturbed social interactions and relationships that could affect their future careers [11].

It has also been reported that extensive use of Facebook is related to behavioral disturbances and poor academic performance among university students, which influences their ways of life and interactions with others [13]. On the other hand, Facebook addiction has been directly linked to anxiety and depression among university students and has impacted their social lives and mental well-being [14].

Nowadays, the extensive use of social media could cause substantial disruptions in the academic achievements of university students. Hence, knowing the magnitudes of facebook addiction and highlighting its predictors is so vibrant in forwarding the ways to challenge this problem. However, no study had examined Facebook addiction and its correlates in Ethiopia as far as we could reach, and this study was considered a pioneer in Ethiopia. Therefore, the main aim of this study was to evaluate the extent of Facebook addiction and its relation to the academic performances of regular undergraduate university students.

In the current study, multiple variables were assessed by standardized tools, which makes its findings sounder. In different previous studies, various variables were evaluated in relation to Facebook addiction. The combination of these different variables in the current study makes it unique.

### Contribution of the current study

Considering this purpose, the findings from this study will contribute in serving as baseline for future studies. Additionally, it will add value to existing knowledge and help provide evidence-based practices. Furthermore, the results of this study will help planners and policymakers in the context of university education.

## Methods and materials

### Participants and study setting

The study was conducted at Mettu University's College of Health Sciences among 422 undergraduate students. Mettu University is one of the public universities found in the southwest of Ethiopia, about 600 km away from the capital city of the country.

### Study period and design

The current study utilized a cross-sectional design and was conducted from December 1–30, 2021.

### Eligibility criteria

Those students who enrolled in the regular program and were active Facebook users were included in the study. Daily active Facebook users were those who logged in at least once per day via the mobile app or a web or mobile browser [15]. First-year students were excluded because of the current Ethiopian educational roadmap [16], where first-year students are not placed in a specific department but stay on common courses until they reach their second year.

### Sample size determination and sampling procedures

The required sample was estimated using the single population formula, considering a 95% confidence interval (CI), a 5% margin of error, and a 50% proportion of occurrence. Consequently, 422 samples were obtained after consideration of an additional 10% nonresponse rate, which was proportionally allocated to each included batch using Bowley's proportional formula ($nh = n\frac{Nh}{N}$) [17], where;

 **nh**–sub-sample from each batch
 **n**–The final sample size of the study = 422.
 **Nh**–The total number of students in each batch
 **N**–The total number of students in the college (source population) = 883.

 After proportional sub-samples were calculated, the required number of participants from each batch was recruited using a systematic random sampling technique, considering the "K" value, which was computed depending on the registration number of the students (Table 1).

### Data collection instruments and procedures

Structured and pretested original English questionnaires were administered to the participants. The questionnaire contained socio-demographic information and questions to assess the status of Facebook addiction, anxiety, depression, and study habits of the study participants.

**Table 1. Sampling procedures used to recruit the study participants.**

| Year level | Nh | Nh | K |
|---|---|---|---|
| 2nd | 298 | 422*298/883 = 142 | 298/142 = 2.10 ≈ 2 |
| 3rd | 295 | 422*295/883 = 141 | 295/141 = 2.10 ≈ 2 |
| 4th | 290 | 422*290/883 = 139 | 290/139 = 2.10 ≈ 2 |

Nh—Total number of the students in each batch, nh—Sample from each batch

K–Interval to include study unit from study population

Demographic characteristics such as age, sex, academic year, and grade point average (GPA) of the students were collected. Facebook addiction was considered an outcome variable and was examined with the Bergen Facebook Addiction Scale (BFAS). The tool was developed by Andreassen et al. and was constructed on six essential elements of addiction (mood modification, silence, conflict, tolerance, withdrawal, and relapse) [18]. The tool had six self-report items that corresponded to each basic component of addiction and were scored on a Likert scale of 1 (very rarely) to 5 (very often) [19]. The tool yields a score of 6–30, in which a higher score indicates greater addiction to Facebook, and the cut-off point for Facebook addiction was suggested by authors as a score ≥ 3 on at least four items (polythetic scoring) [18]. The tool has been widely validated [20–23] and Cronbach's alpha was 0.91 in this study.

The Rosenberg Self-Esteem Scale (RSES) was employed to assess the global self-esteem of the participants. This tool had 10 items, out of which 5 were stated negatively (items 2, 5, 6, 8, and 9) and reversely scored [24]. For items 1, 3, 4, 7, and 10, the tool scored on a 4-point scale ranging from strongly disagree (0) to strongly agree (3), and the inverse for the remaining items. Accordingly, the higher the score, the greater the self-esteem [25]. The tool was widely validated [25–27] and has excellent internal consistency in the current study (CA = 0.95).

The occurrence of depression and anxiety symptoms was examined by the Hospital Anxiety and Depression Scale (HADS). This tool contained seven items for each sub-scale, which were scored on a scale of 0–3 points [28], and it was previously validated in Ethiopia [29]. The Cronbach's alpha values were 0.79 and 0.84, respectively, for the depression and anxiety subscales, and the higher score indicates a higher level of anxiety and depression symptoms.

Study habits were examined by the Study Habit Questionnaire (SHQ) developed by Thomas et al. [30]. The tool had 12 items that were worded positively and scored on a Likert scale of 4, from 1 (strongly disagree) to 4 (strongly agree). The Cronbach's alphas were 0.81 [30] and 0.90 in the original and current studies, respectively.

### Statistical analyses

For all analyses, SPSS version 23.0 (IBM, Armonk, NY, USA) was used. To present categorical variables, percentages and frequencies were employed, while mean and standard deviation (SD) were considered for continuous variables. Analysis of variance (ANOVA) and t-test analyses were used to compute the groups of variables with normal distributions. For post-hoc group analysis, the Tukey HSD test was performed. Multiple regression analysis was performed, all the required assumptions were checked, and no violations were detected. The variance inflation factor (VIF) was used to test for multicollinearity, and no significant collinearity was found. To determine residual independence, the Durbin-Watson test was used, and statistical significance was declared at a p-value of less than 0.05 and a 95% CI.

### Ethical approval and informed consent

All participants had signed written consent before data collection, and all information from participants was kept confidential. An ethical clearance letter (reference number: RPG/100/14) was obtained from the ethical review committee of the College of Health Sciences at Mettu University, and the Helsinki Declaration principles were followed to perform the study.

### Results

### Sociodemographic characteristics of participants

The data from four hundred and three study subjects were fully analyzed, giving a response rate of 95.5%. The participants had a mean age of 23.62 (SD = ±1.79), and 51.6% of them were

male. On average, students spend more than an hour (66.97 minutes ± 48.24) daily using Facebook; 50–400 Ethiopian Birr (ETB) were spent monthly for Facebook use, and 42.9% of participants spent ≥ mean value (Table 2).

The independent samples t-test and one-way ANOVA were used to compare Facebook addiction to the mean score of the groups by sex and academic year, respectively, and no significant differences were found among the groups.

## Psychosocial characteristics of study participants

The participants in the current study had a mean self-esteem score of 14.74 (8.23), and the mean scores for anxiety and depression symptoms indicated an abnormal (case) level for the study subjects, as shown in Table 3.

## The pattern of facebook use among study participants

The mean score of the BFAS was 16.47 (SD = 5.95), indicating that the average number of study participants were addicted to Facebook. As per the suggested cut-off points (score ≥ 3 at least four items), the majority (67.2%) of the students were addicted to Facebook.

## Factors associated with Facebook addiction

Bivariate and multivariable regression analyses were done to detect the predictors of Facebook addiction among students. In the bivariate analysis, anxiety and depressive symptoms showed a positive association with Facebook addiction, whereas the last semester's GPA, study habits, and self-esteem showed a negative correlation with Facebook addiction at a significant level (Table 4).

In the multiple linear regression analysis, the last semester's GPA [β: -10.01, 95% CI: -10.85, -(-9.18)], self-esteem [β: -0.091, 95% CI: -0.139, -(-0.044)], anxiety symptoms (β: 0.104, 95% CI: 0.024–0.183), depressive symptoms [β: 0.026, 95% CI: 0.06–0.112], and study habits [β: -0.008, 95% CI: -0.041, -(-0.056)] showed a statistically significant association with Facebook addiction. In the final model, these predictors contributed a total of 69% of the variance in Facebook addiction among university students (R = 0.829, $R^2$ = 0.687, F = 174.08, P <0.001) (Table 5).

In the current study, there was a significant negative correlation between the GPA of the students and Facebook addiction, in which one unit increase in the student's GPA from the

**Table 2. Characteristics of study participants (n = 403).**

| Variables | Categories | Frequency (%) | Frequency within facebook addiction | |
|---|---|---|---|---|
| | | | Addicted (%) | Non-addicted (%) |
| Sex | Male | 208 (51.6) | 63.9 | 36.1 |
| | Female | 195 (48.4) | 70.8 | 29.2 |
| Academic year | Second | 134 (33.3) | 67.2 | 32.8 |
| | Third | 136 (33.7) | 64.7 | 35.3 |
| | Fourth | 133 (33.0) | 69.9 | 30.1 |
| **Variables** | | **Mean ± SD** | | |
| Age | | 23.62 ± 1.79 | | |
| Time spent (in minutes) per day in using facebook | | 66.97± 48.24 | | |
| Money spent* (ETB) monthly for facebook use | | 174.64 ± 99.89 | | |
| Grade Point Average (GPA) of the last semester | | 3.19 ± 0.44 | | |

**Note**: **ETB**–Ethiopian Birr **SD**—Standard Deviation

*The amount of ETB that need for the purpose of buying data to utilize for facebook

**Table 3. Mean distribution for different psychosocial characteristics of study respondents.**

| Variables | Mean (± SD) | Minimum | Maximum |
|---|---|---|---|
| Self-esteem | 14.74 (± 8.23) | 3.00 | 27.00 |
| Anxiety symptoms | 13.09 (± 5.47) | 1.00 | 20.00 |
| Depression symptoms | 13.18 (± 4.97) | 1.00 | 20.00 |
| Study habits | 29.99 (± 8.53) | 17.00 | 43.00 |

**Note**: SD—Standard Deviation

last semester decreased Facebook addiction by 10.01 (p<0.001). Similarly, as the mean score of students' self-esteem increased by one unit, the Facebook addiction decreased by 0.091 (p<0.001). On the other hand, a point increase in the mean score of anxiety and depressive symptoms, respectively, resulted in a 0.104 and 0.026 (p<0.05) unit increase in the total score of the Facebook addiction scale among university students.

## Discussion

Due to the rising trends of social media usage among university students, Facebook addiction has been examined in many countries. However, this study, which identified Facebook addiction and its correlation to academic performance and other psychosocial variables, was assumed to be the first of its kind in our country. As the study revealed, 67.2% (95% CI = [62.3–72]) of the students were addicted to Facebook. This finding appeared higher than in the previous studies [13, 31–33]. The difference in the findings probably resulted from the difference in the study settings and the year of the studies. On the other hand, because the current study was conducted recently, a higher level of Facebook addiction is expected as evidence indicating the increasing use of social media in the current era [34].

The study found that Facebook addiction has negative relationship with academic performance of the students, as indicated by the last semester's GPA report [β: -10.01, 95% CI: -10.85,-(-9.18)]. This finding is in agreement with the previous studies conducted in India [11], Iraq [9], Pakistan [35], and Sri Lanka [10]. Although various studies have found that Facebook addiction has negative effects on university students, a study from Pakistan [4] found that Facebook could help with communication and information gathering. In another study, Saleem et al. [14] reported the absence of a correlation between Facebook addiction and the academic performance of the students. The discrepancies among these findings might be due to the result of the parameters used to measure academic performance and the tool used to assess the participants. For instance, in the study that reported the usefulness of Facebook use among students, they employed the qualitative (in-depth interview and focus group discussion) means of data collection, in which the drawing of accurate and reliable data is difficult

**Table 4. Bivariable linear regressions for facebook addiction among study participants.**

| Variables | $R^2$ | B | 95% CI | P-value |
|---|---|---|---|---|
| GPA of the last semester | 0.657 | -10.87 | -11.65 –(-10.12) | 0.000 |
| Self-esteem | 0.130 | -0.261 | -0.327 –(-0.194) | 0.000 |
| Anxiety | 0.126 | 0.386 | 0.286–0.485 | 0.000 |
| Depression | 0.108 | 0.393 | 0.282–0.504 | 0.000 |
| Study habit | 0.157 | -.276 | -0.339 –[-0.213] | 0.000 |

**Note**: GPA–Grade Point Average

Table 5. Multivariable linear regression model for facebook addiction among students.

| Variable | Unstandardized coefficients | P-value at 95% CI | 95% CI for β | |
|---|---|---|---|---|
| | | | Lower | Upper |
| Constant | 47.81 | 0.000 | 44.73 | 50.89 |
| GPA of the last semester | -10.01 | 0.000 | -10.85 | -9.18 |
| Self-esteem | -0.091 | 0.000 | -0.139 | -0.044 |
| Anxiety | 0.104 | 0.029 | 0.024 | 0.183 |
| Depression | 0.026 | 0.006 | -0.06 | 0.112 |
| Study habit | -0.008 | 0.000 | -.041 | -.056 |
| **Final Regression Model** | | | | |
| R | R² | Adjusted R² | F | P-value at 95% CI |
| 0.829 | 0.687 | 0.683 | 174.08 | <0.001 |

[36]. Moreover, in the later study, the authors considered the previous year's GPA to measure academic performance, which may mask the real effects of current Facebook addiction.

A significant negative relationship between Facebook addiction and self-esteem was discovered in the current study [β: -0.091, 95% CI: -0.139,-(0.044)] and a positive correlation with the academic achievements of the students. This finding is consistent with the study conducted in Malaysia [37], where non-addicted students had reported higher self-esteem and better academic performance. The finding seems logical, as individuals with high self-esteem are more confident and likely to perform well. A supportive finding has been reported by Blachnio et al. [38], in which individuals with Facebook addiction had lower self-esteem and poorer life satisfaction. Similarly, the study conducted in Iran [39] revealed that lower self-esteem predicted an increase in Facebook addiction among university students.

In the current study, the scores of anxiety and depression symptoms showed a statistically significant positive relationship with Facebook addiction. This finding is supported by a prior study [31], in which up to 20% of Facebook-addicted students reported anxiety and depressive symptoms. Similarly, the study conducted among Pakistani students [11] revealed a strong positive relationship between Facebook addiction and the symptoms of anxiety and depression. Furthermore, we identified a number of previous studies [32, 39–42] that reported the negative correlation of Facebook addiction with anxiety and depressive symptoms in university students.

Even though empirical findings were reached during this study, some limitations are inevitable. For instance, the cross-sectional nature of the current study could limit the cause-effect inference between the outcome variable and its predictors. On the other hand, socio-demographic factors such as the living situation of the students and psychosocial factors were not included. Additionally, only the pattern of Facebook use was examined without consideration of other confounding social media addictions. Lastly, as this was a developing research area in Ethiopia, robust data were not available to compare and contrast the findings.

In spite of these limitations, the results of this study pointed out some imperative findings about the studied problem. Therefore, the current findings could pave the way for any concerned researcher to carry out a future study with a more sophisticated design and to deduce the causal ability of the included and other predictors. The study's utilization of standardized, validated, and widely used tools was considered a strength.

## Conclusion

Although it is considered a major tool of communication, extended use of Facebook causes addiction, which was found to negatively affect the academic performances and mental well-being of the students. To promote the safe and healthy use of Facebook among university

students, appropriate behavioral interventions are crucial. To ensure this, it is better for the legislative body of the university to forward a firm policy to control such sites in the compound to overcome their detrimental effects. Moreover, culturally accepted, adolescent-friendly psychosocial interventions are important for the prevention and management of the problem.

## Supporting information

**S1 File.**
(SAV)

## Acknowledgments

We are grateful to all the data collectors and study participants.

## Author Contributions

**Conceptualization:** Aman Dule, Zakir Abdu, Mohammedamin Hajure, Mustefa Mohammedhussein, Million Girma, Wubishet Gezimu, Abdissa Duguma.

**Data curation:** Aman Dule, Zakir Abdu, Mohammedamin Hajure, Mustefa Mohammedhussein, Million Girma, Wubishet Gezimu, Abdissa Duguma.

**Formal analysis:** Aman Dule, Zakir Abdu, Mohammedamin Hajure, Mustefa Mohammedhussein, Million Girma, Wubishet Gezimu, Abdissa Duguma.

**Funding acquisition:** Aman Dule, Zakir Abdu, Mohammedamin Hajure, Mustefa Mohammedhussein, Million Girma, Wubishet Gezimu, Abdissa Duguma.

**Investigation:** Aman Dule, Zakir Abdu, Mohammedamin Hajure, Mustefa Mohammedhussein, Million Girma, Wubishet Gezimu, Abdissa Duguma.

**Methodology:** Aman Dule, Zakir Abdu, Mohammedamin Hajure, Mustefa Mohammedhussein, Million Girma, Wubishet Gezimu, Abdissa Duguma.

**Project administration:** Aman Dule, Zakir Abdu, Mohammedamin Hajure, Mustefa Mohammedhussein, Million Girma, Wubishet Gezimu, Abdissa Duguma.

**Resources:** Aman Dule, Zakir Abdu, Mohammedamin Hajure, Mustefa Mohammedhussein, Million Girma, Wubishet Gezimu, Abdissa Duguma.

**Software:** Aman Dule, Zakir Abdu, Mohammedamin Hajure, Mustefa Mohammedhussein, Million Girma, Wubishet Gezimu, Abdissa Duguma.

**Supervision:** Aman Dule, Zakir Abdu, Mohammedamin Hajure, Mustefa Mohammedhussein, Million Girma, Wubishet Gezimu, Abdissa Duguma.

**Validation:** Aman Dule, Zakir Abdu, Mohammedamin Hajure, Mustefa Mohammedhussein, Million Girma, Wubishet Gezimu, Abdissa Duguma.

**Visualization:** Aman Dule, Zakir Abdu, Mohammedamin Hajure, Mustefa-Mohammedhussein, Million Girma, Wubishet Gezimu, Abdissa Duguma.

**Writing – original draft:** Aman Dule, Zakir Abdu, Mohammedamin Hajure, Mustefa Mohammedhussein, Million Girma, Wubishet Gezimu, Abdissa Duguma.

**Writing – review & editing:** Aman Dule, Zakir Abdu, Mohammedamin Hajure, Mustefa Mohammedhussein, Million Girma, Wubishet Gezimu, Abdissa Duguma.

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
