## [Decision Letter · Decision Letter 0]

19 Jul 2022

PONE-D-22-11967Facebook addiction and affected academic performance among Ethiopian university students: A cross-sectional StudyPLOS ONE

Dear Dr. Dule,

Thank you for submitting your manuscript to PLOS ONE. After careful consideration, we feel that it has merit but does not fully meet PLOS ONE’s publication criteria as it currently stands. Therefore, we invite you to submit a revised version of the manuscript that addresses the points raised during the review process.

We look forward to receiving your revised manuscript.

Kind regards,

Md. Tanvir Hossain

Academic Editor

PLOS ONE

Journal Requirements:

2. PLOS ONE does not copy edit accepted manuscripts (https://journals.plos.org/plosone/s/criteria-for-publication#loc-5). To that effect, please ensure that your submission is free of typos and grammatical errors

Reviewers' comments:

Reviewer's Responses to Questions

**Comments to the Author**

1. Is the manuscript technically sound, and do the data support the conclusions?

Reviewer #1: No

Reviewer #2: No

Reviewer #3: Yes

2. Has the statistical analysis been performed appropriately and rigorously? 

Reviewer #1: N/A

Reviewer #2: Yes

Reviewer #3: No

3. Have the authors made all data underlying the findings in their manuscript fully available?

Reviewer #1: Yes

Reviewer #2: Yes

Reviewer #3: Yes

4. Is the manuscript presented in an intelligible fashion and written in standard English?

Reviewer #1: Yes

Reviewer #2: No

Reviewer #3: Yes

5. Review Comments to the Author

Reviewer #1: 1 - The abstract is long, and the authors should shorten it.

2 - Bringing the research results in the abstract is not compatible with the structure of the one standard paper, and the authors should correct it.

3- Authors should explain the reasons for doing this research and the current challenges in the introduction section.

4 - Authors should add the Contribution section at the end of the introduction.

5 - Why there aren't Methodology and Evaluation sections in the manuscript?

Reviewer #2: I am very grateful to the Editor for giving me a chance to review the manuscript entitled “Facebook addiction and affected academic performance among Ethiopian university students: A cross-sectional Study” to the journal of “PLOS ONE”. The main objective of this study was to evaluate the status of facebook addiction and its relation with academic performance and other correlates among university students. I appreciate the time and effort that the authors have dedicated to preparing the manuscript. I read the manuscript very carefully and consequently raised the quarries and suggestions under two headings: major revisions and minor revisions to improve the manuscript which are as follows:

Major Revisions:

1. In introduction a clear explanation is needed why this study is different from other studies.

2. Major revisions are required in methods and materials section particularly, participants and study settings, study periods and design, illegibility criteria, sample size determination and sampling procedures points. Authors have to give logical explanation under each point for instance, under illegibility criteria, authors mentioned that those students who enrolled into regular program and active facebook users were included in the study; while the first year students were excluded. How did they measure active facebook users? As well as why did they exclude first year students? So logical explanation is needed behind excluding first year students in this study.

3. Logical explanation is needed why this study consider university students as respondents under participants and study settings point of methods and material section.

4. Under sample size determination and sampling procedures point, the authors mentioned that systematic random sampling was used to select the sample size. But the authors did not mention the sampling procedures in details like, what is the starting point or sample number and how did they select that sample number and so on.

5. The conclusion is not presented logically supported by the obtained results.

6. Under Ethical approval and informed consent point, the authors mentioned that the ethical clearance was provided by the ethical review committee of college of health sciences, Mettu University and the Helsinki Declaration principles was kept to perform the study (line no. 251-253). However, they did not mention the reference number of ethical approval.

7. The reference section have to revise (reference no. 5, 9, 11, 23, 27, 30) following the guideline of the journal ‘PLOS ONE’. Some references are very old such as reference no. 19 and 20 are in 2001 and 2007. Recent research works have to incorporate.

8. Overall, the writing of the manuscript have to improve following Standard English.

Minor Revisions:

1. Revision is needed in Key words such as, Self-esteem, Anxiety and depression, Facebook addiction, Study habits and University students (Line no. 48-50).

2. In Introduction section (line no. 56-57) have to rewrite.

3. In conclusion (line no. 235-237) As the results indicated, facebook addiction is affecting mental wellbeing of the students and self-esteem has identified as negative predictor of facebook addiction. This sentences have to rewrite.

The author(s) are suggested to rewrite the manuscript accordingly. Hopefully, the aforesaid comments and suggestions would help to enrich this manuscript.

Reviewer #3: “Sheet for Comments”

Overall for the whole paper please see the format of writing for the journal and format your paper.

Abstract

1. Please write the name of the questionnaires sequentially as per your variables addressed in the study.

2. The long descriptions on statistical analyses run for the study is not necessary in abstract rather some values with the main findings are appreciable.

3. The information on ethics needs a reference number.

Introduction

The introduction is written well and concise.

Methods

1. Ethics information is needed to be included in the methodology section.

2. What demographic variables have you collected?? It is needed to have description on the demographic variables. For instance, different demographic information on the students, like which year, subjects, etc.

3. It would be great if you give a consort flow chart on the inclusion of the participants.

4. Please write about the questionnaires used in your study in the way the journal asks. For instance, subscales? How many?? What is Likert of 1?? What is CA (Chronbach alpha)?? Please state Chronbach Alpha for all the scales.

5. In the statistical analyses section it is needed to be described clearly about the how the multi-collinearity of the variables were assessed and how they were managed. How the correlated variables were managed in analyses for the regression.

Result:

1. Descriptive statistics are needed for the Demographic variables in relation to the independent variables in Table 2 (minimum and maximum are not needed)

2. Some of the descriptions are not clearly stated, like, what do you wanted to mean by money spent for FB? Is it money spent for buying data for FB or else?

3. The demographic information given in Table 1 will be better fitted in sample / participant section.

4. The description of the regression analysis is repeated on the result section and conclusion as well.

Conclusions

1. It is needed to be written without repeating the results in the conclusions again.

Reference

1. Many of the references are not in correct format, for instance they are in all capital letters’ in references

6. PLOS authors have the option to publish the peer review history of their article (what does this mean?). If published, this will include your full peer review and any attached files.

Reviewer #1: **Yes: **Saman Forouzandeh

Reviewer #2: No

Reviewer #3: No

---

## [Author Response · Author response to Decision Letter 0]

2 Nov 2022

Response to Reviewers

First of all, we are very glad to get the soon response which could initiates the authors to work with you even in the future. We would like also to appreciate the reviewers for their valuable and useful feedback which guides us to look the weak parts of our work and improve it accordingly. Saying this, we had tried to correct all the comments and suggestions from the editor and each reviewer point-by-point to the extent that we assumed the response could satisfy the reviewers as appeared below and had highlighted the changes in the manuscript.

Review Comments to the Author

Reviewer #1: 

1 - The abstract is long, and the authors should shorten it.

Response 1 – We have found this suggestion useful and the manuscript has shorten accordingly 

2 - Bringing the research results in the abstract is not compatible with the structure of the one standard paper, and the authors should correct it.

Response 2 – We had accepted the suggestion and some explanation of the result in manuscript section has been removed

3- Authors should explain the reasons for doing this research and the current challenges in the introduction section.

Response 3 – Accordingly, The logical explanation has been given why we conducted this study and the current challenges has been highlighted in introduction section (Line number 55 - 59 of the revised manuscript)

4 - Authors should add the Contribution section at the end of the introduction.

Response 4 – This section has added as per suggestion (Line number 88 - 92)

5 - Why there aren't Methodology and Evaluation sections in the manuscript?

Response 5 – As per the journal guideline for manuscript formatting, the manuscript has “Methods and Materials” section and we thought that is enough to include all utilized methods and procedures

Reviewer #2:

I am very grateful to the Editor for giving me a chance to review the manuscript entitled “Facebook addiction and affected academic performance among Ethiopian university students: A cross-sectional Study” to the journal of “PLOS ONE”. The main objective of this study was to evaluate the status of facebook addiction and its relation with academic performance and other correlates among university students. I appreciate the time and effort that the authors have dedicated to preparing the manuscript. I read the manuscript very carefully and consequently raised the quarries and suggestions under two headings: major revisions and minor revisions to improve the manuscript which are as follows:

Major Revisions:

1. In introduction a clear explanation is needed why this study is different from other studies.

Response 1 – In general, the underlying importance of this study was that it has considered the first one nationally and its contribution is expected valuable. In addition to this, we had included different variables in the way we considered unique from previous studies in countries other than Ethiopia. We had added logical explanation about this concern in the introduction section of the revised manuscript (Line number 84 - 87)

2. Major revisions are required in methods and materials section particularly, participants and study settings, study periods and design, illegibility criteria, sample size determination and sampling procedures points. Authors have to give logical explanation under each point for instance, under illegibility criteria, authors mentioned that those students who enrolled into regular program and active facebook users were included in the study; while the first year students were excluded. How did they measure active facebook users? As well as why did they exclude first year students? So logical explanation is needed behind excluding first year students in this study.

Response 2 – We would like to thank the reviewer for this important suggestion. As we found it very crucial, these points have been addressed in revised manuscript (Line number 102 - 106)

3. Logical explanation is needed why this study consider university students as respondents under participants and study settings point of methods and material section.

Response 3 – We considered university students as respondents for this specific problem after careful review of previous materials. While we identified this problem as a research topic, we had found that currently university and college going students were the major victims. On the other hands, as the instructors of higher education institutions, authors are facing many challenges with the academic achievements of their students and they found the university students as important population specially for problem related to emerging technologies. The current challenges concerning this population in relation to the problem under study has been explained in the manuscript also (Line number 55 - 59).

4. Under sample size determination and sampling procedures point, the authors mentioned that systematic random sampling was used to select the sample size. But the authors did not mention the sampling procedures in details like, what is the starting point or sample number and how did they select that sample number and so on.

Response 4 – We had found this suggestion useful and we had added the detail of sampling procedures and participant inclusion (Line number 111 - 123 in revised manuscript)

5. The conclusion is not presented logically supported by the obtained results.

Response 5 – The conclusion has been modified as per the suggestion (Line number 269 - 272)

6. Under Ethical approval and informed consent point, the authors mentioned that the ethical clearance was provided by the ethical review committee of college of health sciences, Mettu University and the Helsinki Declaration principles was kept to perform the study (line no. 251-253). However, they did not mention the reference number of ethical approval.

Response 6 – Reference number has been added as per the suggestion (Line number 166 - 167)

7. The reference section have to revise (reference no. 5, 9, 11, 23, 27, 30) following the guideline of the journal ‘PLOS ONE’. Some references are very old such as reference no. 19 and 20 are in 2001 and 2007. Recent research works have to incorporate.

Response 7 – All suggested references has been corrected except reference number 19 and 20 (in first manuscript), but reference number 24 and 25 in the revised manuscript. As this references are indicating the validation of the tool, we didn’t replaced them with other references

8. Overall, the writing of the manuscript have to improve following Standard English.

Response 8 – The whole manuscript has been revised for standard English by the expert as per suggestion

Minor Revisions:

1. Revision is needed in Key words such as, Self-esteem, Anxiety and depression, Facebook addiction, Study habits and University students (Line no. 48-50).

Response 1 – some modification has been made to the keywords in revised manuscript (Line number 47 - 48)

2. In Introduction section (line no. 56-57) have to rewrite.

Response 2 – This has been rewritten as per the suggestion 

3. In conclusion (line no. 235-237) As the results indicated, facebook addiction is affecting mental wellbeing of the students and self-esteem has identified as negative predictor of facebook addiction. This sentences have to rewrite.

Response 3 – Over all, the conclusion has been modified and rewritten.

The author(s) are suggested to rewrite the manuscript accordingly. Hopefully, the aforesaid comments and suggestions would help to enrich this manuscript.

Reviewer #3: 

“Sheet for Comments”

Overall for the whole paper please see the format of writing for the journal and format your paper.

Response – This suggestion has been accepted and the manuscript has reviewed accordingly

Abstract

1. Please write the name of the questionnaires sequentially as per your variables addressed in the study.

Response 1 – We had just explained the included tools in their order of appearance in the revised manuscript

2. The long descriptions on statistical analyses run for the study is not necessary in abstract rather some values with the main findings are appreciable.

Response 2 – We found this as useful suggestion and we had modified the abstract accordingly

3. The information on ethics needs a reference number.

Response 3 – The reference number has been added in the revised manuscript (line number 166 - 167)

Introduction

The introduction is written well and concise. - Thank you very much for your constructive idea.

Methods

1. Ethics information is needed to be included in the methodology section.

Response 1 – We had shifted the ethical information in to the methods of the manuscript (Line number 163 - 167)

2. What demographic variables have you collected?? It is needed to have description on the demographic variables. For instance, different demographic information on the students, like which year, subjects, etc.

Response 2 – The specific demographic variables has been explained in the revised manuscript (Line number 128 - 129)

3. It would be great if you give a consort flow chart on the inclusion of the participants.

Response 3 – As per the suggestion, the details of participant inclusion with flow chart has been added to revised manuscript (Line number 111 - 123)

4. Please write about the questionnaires used in your study in the way the journal asks. For instance, subscales? How many?? What is Likert of 1?? What is CA (Chronbach alpha)?? Please state Chronbach Alpha for all the scales.

Response 4 – we had tried to explain all scales with respective sub-scales and, for all tools, the Cronbach’s alpha has been explained

5. In the statistical analyses section it is needed to be described clearly about the how the multi-collinearity of the variables were assessed and how they were managed. How the correlated variables were managed in analyses for the regression.

Response 5 – This point has explained well in revised manuscript (Line number 159 - 161)

Result:

1. Descriptive statistics are needed for the Demographic variables in relation to the independent variables in Table 2 (minimum and maximum are not needed)

Response 1 – This table has modified as per suggestion and the descriptive value in respect to dependent variable has also given in revised manuscript (Line number 75, table 2)

2. Some of the descriptions are not clearly stated, like, what do you wanted to mean by money spent for FB? Is it money spent for buying data for FB or else?

Response 2 – This explained accordingly in revised manuscript (line number 177)

3. The demographic information given in Table 1 will be better fitted in sample / participant section.

Response 3 – From this suggestion, we thought as the information included in this table should explained in method section and we had added that in socio-demographic characteristics of the participants in method (Line number 128 - 129).

4. The description of the regression analysis is repeated on the result section and conclusion as well.

Response 4 – This point has addressed in revised manuscript as we made the modification to conclusion part (Line number 169 - 172)

Conclusions

1. It is needed to be written without repeating the results in the conclusions again.

Response 1 – Rewritten accordingly (Line number 169 - 1712)

Reference

1. Many of the references are not in correct format, for instance they are in all capital letters’ in references

Response 1 – All references has been reviewed and necessary correction has made in revised manuscript (Line number 286)

---

## [Decision Letter · Decision Letter 1]

12 Dec 2022

PONE-D-22-11967R1Facebook addiction and affected academic performance among Ethiopian university students: A cross-sectional StudyPLOS ONE

Dear Dr. Dule,

Thank you for submitting your manuscript to PLOS ONE. After careful consideration, we feel that it has satisfied our scientific requirements for publication.

However, our editorial team have significant concerns about the grammar, usage, and overall readability of the manuscript. PLOS ONE requires that published manuscripts use language which is 'clear, correct, and unambiguous', see our criteria for publication at https://journals.plos.org/plosone/s/criteria-for-publication#loc-5. We therefore request that you revise the text to fix the grammatical errors and improve the overall readability of the text.

We suggest you have a fluent English-language speaker thoroughly copyedit your manuscript for language usage, spelling, and grammar. If you do not know anyone who can do this, you may wish to consider employing a professional scientific editing service.

Whilst you may use any professional scientific editing service of your choice, PLOS has partnered with both American Journal Experts (AJE) and Editage to provide discounted services to PLOS authors. Both organizations have experience helping authors meet PLOS guidelines and can provide language editing, translation, manuscript formatting, and figure formatting to ensure your manuscript meets our submission guidelines. To take advantage of our partnership with AJE, visit the AJE website (https://www.aje.com/go/plos/) for a 15% discount off AJE services. To take advantage of our partnership with Editage, visit the Editage website (www.editage.com) and enter referral code PLOSEDIT for a 15% discount off Editage services. If the PLOS editorial team finds any language issues in text that either AJE or Editage has edited, the service provider will re-edit the text for free.

Please note that we will not be able to proceed with publication of your manuscript until the concerns above are addressed.

* A copy of your manuscript showing your changes by either highlighting them or using track changes (uploaded as a supporting information file)

* A clean copy of the edited manuscript (uploaded as the new manuscript file)

We look forward to receiving your revised manuscript.

Kind regards,

Steve Zimmerman, PhD

Associate Editor, PLOS ONE

on behalf of

Tanvir Hossain

Academic Editor

PLOS ONE

We look forward to receiving your revised manuscript.

Kind regards,

Md. Tanvir Hossain

Academic Editor

PLOS ONE

Journal Requirements:

Reviewers' comments:

Reviewer's Responses to Questions

**Comments to the Author**

1. If the authors have adequately addressed your comments raised in a previous round of review and you feel that this manuscript is now acceptable for publication, you may indicate that here to bypass the “Comments to the Author” section, enter your conflict of interest statement in the “Confidential to Editor” section, and submit your "Accept" recommendation.

Reviewer #1: All comments have been addressed

Reviewer #2: All comments have been addressed

Reviewer #3: All comments have been addressed

2. Is the manuscript technically sound, and do the data support the conclusions?

Reviewer #1: No

Reviewer #2: Yes

Reviewer #3: Yes

3. Has the statistical analysis been performed appropriately and rigorously? 

Reviewer #1: No

Reviewer #2: Yes

Reviewer #3: Yes

4. Have the authors made all data underlying the findings in their manuscript fully available?

Reviewer #1: No

Reviewer #2: Yes

Reviewer #3: Yes

5. Is the manuscript presented in an intelligible fashion and written in standard English?

Reviewer #1: Yes

Reviewer #2: Yes

Reviewer #3: Yes

6. Review Comments to the Author

Reviewer #1: I rejected the paper the previous time. I rejected the paper the previous time. I rejected the paper the previous time. I rejected the paper the previous time.

Reviewer #2: (No Response)

Reviewer #3: The paper has been significantly improvised.

A big congratulations to the authors.

It is ready to meet the press after rigorus proof reading.

Thank you

7. PLOS authors have the option to publish the peer review history of their article (what does this mean?). If published, this will include your full peer review and any attached files.

Reviewer #1: No

Reviewer #2: No

Reviewer #3: No

---

## [Author Response · Author response to Decision Letter 1]

24 Dec 2022

Thanks to all reviewers, they all had reported as all previously raised comments has addressed, so we didn’t get any comments at this stage except the editors’ concern of the English language, grammar usage and overall readability of the manuscript and suggested edition. Accordingly, our manuscript has copy edited for English language by professional English language academic instructor and the details of this professional has attached separately. Concerning the other suggestions like changes to financial disclosure and the references, no change has made to financial disclosure and the references were reviewed thoroughly to fulfill the journal requirement and no retracted article has been referenced. 

Thank you again and we will wait for your positive response!

Best Regards!

---

## [Editor Report · Decision Letter 2]

27 Dec 2022

Facebook addiction and affected academic performance among Ethiopian university students: A cross-sectional Study

PONE-D-22-11967R2

Dear Dr. Dule,

We’re pleased to inform you that your manuscript has been judged scientifically suitable for publication and will be formally accepted for publication once it meets all outstanding technical requirements.

Kind regards,

Md. Tanvir Hossain

Academic Editor

PLOS ONE

---

## [Editor Report · Acceptance letter]

27 Jan 2023

PONE-D-22-11967R2 

Facebook addiction and affected academic performance among Ethiopian university students: A cross-sectional Study 

Dear Dr. Dule:

I'm pleased to inform you that your manuscript has been deemed suitable for publication in PLOS ONE. Congratulations! Your manuscript is now with our production department. 

Kind regards, 

on behalf of

Dr. Md. Tanvir Hossain 

Academic Editor

PLOS ONE